# A Vicarious Technique for Understanding and Diagnosing Hyperspectral Spatial Misregistration

**DOI:** 10.3390/s23094333

**Published:** 2023-04-27

**Authors:** David N. Conran, Emmett J. Ientilucci

**Affiliations:** Chester F. Carlson Center for Imaging Science, Digital Imaging and Remote Sensing Laboratory, Rochester Institute of Technology, 54 Lomb Memorial Drive, Rochester, NY 14623, USA

**Keywords:** SPSF, keystone, spatial misregistration, hyperspectral, multispectral, imaging, small unmanned aircraft systems, UAS, UAV, convex mirrors

## Abstract

Pushbroom hyperspectral imaging (HSI) systems intrinsically measure our surroundings by leveraging 1D spatial imaging, where each pixel contains a unique spectrum of the observed materials. Spatial misregistration is an important property of HSI systems because it defines the spectral integrity of spatial pixels and requires characterization. The IEEE P4001 Standards Association committee has defined laboratory-based methods to test the ultimate limit of HSI systems but negates any impacts from mounting and flying the instruments on airborne platforms such as unmanned aerial vehicles (UAV’s) or drones. Our study was designed to demonstrate a novel vicarious technique using convex mirrors to bridge the gap between laboratory and field-based HSI performance testing with a focus on extracting hyperspectral spatial misregistration. A fast and simple extraction technique is proposed for estimating the sampled Point Spread Function’s width, along with keystone, as a function of wavelength for understanding the key contributors to hyperspectral spatial misregistration. With the ease of deploying convex mirrors, off-axis spatial misregistration is assessed and compared with on-axis behavior, where the best performance is often observed. In addition, convex mirrors provide an easy methodology to exploit ortho-rectification errors related to fixed pushbroom HSI systems, which we will show. The techniques discussed in this study are not limited to drone-based systems but can be easily applied to other airborne or satellite-based systems.

## 1. Introduction and Background

### 1.1. Hyperspectral Imaging

Hyperspectral imaging (HSI) utilizes instrumentation that provides a unique perspective for the simultaneous observation of spatial and spectral scene content. Each spatial pixel contains a unique spectrum that is directly proportional to the physical characteristics of the observed material. This is a major advantage over traditional filter-based imaging systems because the measured spectrum can be used more effectively to detect objects, materials, or changes within the scene. For all the usefulness an HSI system promotes in remote sensing applications, the complex optical design of hyperspectral instruments introduces spatial imperfections on the imaged scene, requiring new and innovative ways to diagnose the impact of spatial artifacts related to spectral-spatial information.

A simple design of a pushbroom HSI system can be seen in Figure 1, with the geometric projection of the slit being one-to-one with the detector pixel pitch in the spectral direction. Generally, a pushbroom HSI system only records spatial information in the cross-track direction, where each pixel is dispersed into a spectrum. The remaining spatial component must then be built up by the movement of the HSI system in the along-track direction. Thus, a 3D image of the scene is created, with each pixel containing a spectrum of the 2D spatial scene.

### 1.2. Imaging Performance and Spatial Misregistration

Imaging performance of hyperspectral (HS) instruments is a complex topic of discussion because of the unique combination of radiometric, spectral, spatial, and temporal artifacts contaminating the imaged scene. Due to the complex nature of HSI systems, the IEEE P4001 Standards Association committee is currently developing guidelines, definitions, and testing procedures to help the HS community better understand how these systems operate [1]. Figure 1 illustrates an ideal HSI system and for a given spatial pixel within the scan line, the recorded spectrum should only originate from a single pixel without any influence from the surrounding area. However, due to diffraction and various aberrations, a perfect spatial mapping with a unique spectrum is not possible and leads to spatial misregistration [2]. Further degradation in spatial performance is introduced by motion blur and imperfect ortho-rectification, including post-processing artifacts such as nearest-neighbor interpolation. To quantify spatial misregistration, a spectral measurement of the sampled Point Spread Function (SPSF) is required and is further recommended by the P4001 Standards Committee [1]. The SPSF is a culmination of all system imperfections, including optical aberrations, diffraction, grating dispersion, detector pixel blurring, and sampling.

Figure 2 illustrates the main artifacts that contribute to spatial misregistration seen within HSI systems and include the following: *keystone*, SPSF width variability, and SPSF shape differences [2]. The top row shows a well-known distortion called *keystone* that describes the displacement of the SPSF’s centroid position as a function of wavelength and can be related back to a variation in the entrance slit magnification when focusing broadband illumination onto the focal plane. The middle row shows a wavelength dependent SPSF width originating from optical diffraction, aberrations, and specifically for HSI systems, the entrance slit and grating diffraction. The last row shows the combination spatial artifacts related to both keystone and SPSF shape differences that are often seen in real HSI systems [3]. What is not shown in Figure 2 are the field-of-view (FOV) dependencies on the various parameters that define spatial misregistration. To fully quantify HSI spatial misregistration, all off-axis behavior is required to be measured as well. In the work presented, we will show that the HSI system under test has degraded performance at the edges of the FOV.

For the reason that spatial misregistration encompasses multiple spectral measurements of the SPSF’s behavior, having a single graph to characterize imaging performance would be more desirable for quick diagnostic checks. To quantify HSI spatial misregistration, a wavelength pair calculation known as *spatial coregistration error* (Equation (Equation 1)) can be used to measure differences in the SPSF’s keystone, width, and shape across the spectral range [4,5]. The assessment of spatial coregistration error relies on the area normalized difference between two SPSF’s for all spectral channel pairs. The spatial coregistration error has a range from 0.0 (i.e., completely overlapping SPSF’s) to 1.0 (i.e., completely disjoint SPSF’s) and will highlight the behavior of the SPSF at various wavelength pairs and at different locations within the imagers FOV. It should be noted that a smaller spatial coregistration error is ideal (e.g., closer to 0.0 is best) and will lead to fewer errors in applications where spatial and spectral integrity are important. Moreover, for sharp SPSF’s, we will show that keystone errors contribute greatly to spatial coregistration error when compared with spatial width variability. The method of understanding spatial misregistration in HSI systems is very important for characterizing the imaging systems spatial performance and has been demonstrated in laboratory experiments using line sources [6].

The calculation of 1D spatial coregistration error ($\epsilon_{i,j}$) for all wavelength pairs i,j is defined as
(1)ϵi,j=12∫|SPSFi(x)−SPSFj(x)|dx
where SPSF is an area normalized Gaussian distribution fitted to point targets integrated over the spatial dimension. It is important to note that this calculation is sensitive to enclosed energy, and under estimating enclosed energy will lead to a better estimate of spatial coregistration error [4]. Equation (Equation 1) can be easily extended into a second dimension, where the volume normalized SPSF is required.

Laboratory experiments tend to be performed in a static setting by scanning a line source over various parts of the focal plane to extract spatial misregistration. However, extracting spatial misregistration during field experiments using ideal point sources has never been discussed, especially for drone-based imaging platforms. This study fills in the gap for vicarious estimates of HSI spatial misregistration during field experiments by using convex mirror targets and a simple algorithm design for approximating the SPSF response with a 2D Gaussian distribution. However, only the cross-track direction will be assessed since this can be directly tied back to the internal performance of HSI systems, which contributes to the overall coregistration error estimate. More importantly, off-axis behavior can be easily characterized by simply allowing the drone to fly over deployed (convex mirrors) targets until they are no longer in the imagers FOV.

### 1.3. HSI Application-Based Errors

Spatial coregistration error not only characterizes the spatial misregistration but can also diagnose the ultimate limit on application-based spectral/spatial measurements with HSI systems. When these unknown spatial distortions are present within the HS instruments, small target signatures become mixed in with the background and resemble a problem similar to spectral unmixing with a relationship to the SPSF misregistration.

For example, if we assume the following about an ideal HS instrument: detector-limited SPSF (i.e., width of the optical PSF is equivalent to detector pixel area) with a wavelength-dependent width that varies linearly and a bright sub-pixel target on top of a uniform background. The sub-pixel target will dispense energy into surrounding background pixels, where the energy loss from the target is proportional to the SPSF shape and size. If a spectral unmixing algorithm were applied to this simplistic scenario, energy from the SPSF tails that bleed into surrounding pixels would have a spectral component related to both the sub-pixel target and background. Since the SPSF shape and size are not consistent across the spectral range (or FOV), this would indicate varying degrees of mixture between target and background. The sub-pixel targets impact is no longer contained to a single spatial pixel and adds to the complexity of identifying sub-pixel targets from a background. This example can be expanded upon by now assuming a realistic HSI system with FOV-dependent SPSF characteristics and added keystone, which results in an even more complicated situation. Thus, using convex mirrors to understand an HSI system’s spatial misregistration under field experiment conditions (i.e., complex motion on drone platforms) can now allow scientists to further understand any issues in their scientific results.

Another important application that can suffer from unknown spatial misregistration is small target radiometry and sub-pixel target detection. Target detection relies on the ability to take known spectral signatures and spatially identify unknown sub-pixel target positions with a similar spectral makeup. Depending on the degree of spatial misregistration, the unknown sub-pixel target spectrum can become contaminated by the background, as discussed above, creating a spectrum that does not physically match the defined target. This has large impacts for spectral signatures that were selected from databases containing material properties measured from non-imaging spectrometers (e.g., hand-held point spectrometers). Large spatial misregistration, especially keystone, would corrupt HSI data such that it would not have comparable spectral signatures to the materials in the database. Furthermore, cluttered backgrounds and significant changes in spatial misregistration across the FOV will exacerbate this issue [7,8].

## 2. Methodology

### 2.1. Field Experiment Overview

In order to demonstrate the usability of convex mirrors to extract spatial misregistration from HSI systems, a variety of field experiments were performed. Additionally, we performed field experiments with an MSI system to support our ability to extract accurate SPSF estimates. In this section, we will highlight all field experiments while noting the ground sampling distance (GSD) and instantaneous field-of-view (IFOV) for each instrument. Specific imaging system manufacturers will not be discussed so that attention will be focused on the use of the technique instead of quantifying an instrument’s performance. All imaging systems were fixed mounted to the underside of a DJI Matrice 600 unmanned aerial system (UAS) or drone.

Figure 3 illustrates the field experiment that was conducted to estimate an MSI systems SPSF, utilizing our oversampling technique. Here, only the blue channel with a center wavelength of 475 nm with a bandwidth of 20 nm and an IFOV of 0.694 mrad was used to demonstrate the use of point targets. The drone was flown at an altitude of 57.6 m resulting in imagery collected at 4.0 cm GSD. In Figure 3, there were 16 similar point targets distributed over two black felt-covered panels (each 1.2 m × 1.2 m in size) with an edge target off to the side for comparison, which will not be discussed in this study.

The primary data set used to analyze the FOV-dependent spatial misregistration of the HSI system was conducted at two locations during the Ground to Space Calibration Experiment (G-SCALE) [9]. G-SCALE was a joint campaign with the National Research Council of Canada (NRC), Labsphere Inc., MAXAR and the Rochester Institute of Technology (RIT). The HSI system has a spectral range of 400 nm–1000 nm and spectral sampling of 2.23 nm with an average spectral resolution of 5 nm and an IFOV of 0.617 mrad. Figure 4 shows the primary testing site for G-SCALE where the drone flew at an altitude of 103 m resulting in imagery collected at 6.5 cm GSD. The primary site was designed for radiometric and spatial testing [10] and comprised of three black felt-covered panels (1.2 m × 1.2 m in size) with three various mirror configurations. Figure 5 was the secondary site where the drone flew at an altitude of 81.0 m resulting in imagery collected at 5.0 cm GSD. The secondary site was designed for additional spatial testing with large spectral unmixing targets showing two mirror configurations. For both sites, multiple overpasses over the targets were collected and used to analyze spatial misregistration at different parts of the imagers FOV. The spectral unmixing targets were deployed for a separate experiment not to be discussed in this paper.

We have been able to extract consistent keystone estimates from a localized region on the focal plane, which demonstrates the repeatability of our presented technique (see the end of Section 3.2). Here, the same HSI system was used as in G-SCALE, but it was flown at an altitude of 63.2 m, resulting in imagery collected at 3.9 cm GSD. Figure 6 shows the deployment area for the various mirror configurations (i.e., single and multiple mirror arrays) on black felt-covered panels. BigMAC was another multi-agency field experiment used to assess the use of HSI systems to validate surface reflectance products for current and future Landsat missions [11]. KBR Wyle, contractor to USGS EROS, South Dakota State University, Rochester Institute of Technology, and Labsphere, Inc. were the main participants in this exercise, where Lambertian panels and convex mirrors were the main field targets.

### 2.2. Sparc Target Overview

To assess spatial misregistration in drone-based HSI systems, the SPecular Array Radiometric Calibration (SPARC) method uses convex mirrors to rely on solar radiation to an imaging system for both radiometric calibration and spatial characterization [12,13,14]. This paper will have a primary focus on using convex mirrors or point targets for spatial characterization. An initial radiometric assessment of point targets for *drone-based HS imagery* can be found in [9,10]. Spatially, the convex mirrors form a virtual image of the solar disk at the focal plane with a solar-like spectral signature creating a bright, sub-pixel point source. The reflected solar radiation has dependencies on the radius of curvature ($R_{m}$) or the focal length ($f_{m}=2\cdot R_{m}$), and the solar disk’s virtual image can be viewed from any angle within the Field-of-Regard (FOR) [13]. The projected area of the convex mirror ($D_{m}$) and $R_{m}$ define the FOR. Mirror configurations used for all experiments can be found in Table 1. Simple geometric projections of the solar’s angular diameter through the convex mirrors can provide estimates of the solar disk’s physical diameter ($d_{sun}$) as seen by an imaging system. Figure 7 illustrates the various geometric relationships for convex mirrors.

During the field experiments discussed in Section 2.1, the GSD’s at least 3.9 cm or larger, and even for the largest mirror configuration (i.e., $R_{m}$ = 100 mm), the ratio of $d_{sun}$ to the GSD is 1.1% and smaller for the other mirrors. Thus, when observing the point targets from the drone-based platforms, the resultant image is the SPSF and defines the end-to-end spatial performance of the imaging systems. This includes all blurring factors such as the optical PSF, entrance slit diffraction, the light-sensitive area of a pixel, all forms of motion blur and, any post-processing errors (i.e., interpolation and ortho-rectification inconsistencies). It can be stated that convex mirrors produce an ideal point source and will be the main technique for estimating the spatial misregistration in HSI systems in this study. Furthermore, the SPARC technique will demonstrate the use of multiple points to provide an oversampling technique for multispectral imaging (MSI) systems for more accurate SPSF estimates where ortho-rectification issues are non-existent when compared with the HSI system. For more stable imaging platforms for HSI imaging (i.e., airborne or satellite-based instruments), the oversampling technique can be applied to a localized region within an HS image [15,16].

When imaging point targets with pushbroom HSI systems, the second spatial dimension (i.e., the along-track) is captured by platform motion, and GPS/IMU data is used to ortho-rectify a final 2D spatial image where each pixel contains a spectrum. When pushbroom HSI systems are fixed mounted to the moving platform (e.g., the HSI system in this study), the scene reconstruction can be plagued with ortho-rectification errors over areas where complex roll and/or pitch can be experienced. The response time on the GPS/IMU will be the ultimate limit on how much roll/pitch can be experienced without causing massive errors in the final image product. For the reason that motion can plague the imagery used in this study and unrealistic reconstruction can be experienced, using the presented oversampling technique to estimate a more accurate SPSF shape will limit the spatial coregistration error to just keystone and SPSF width. Thus, the proposed solution will be to approximate the HS instruments SPSF with a 2D Gaussian distribution and only focus on the cross-track spatial performance. The HSI system’s cross-track performance is related to the instrument’s internal imperfections with small amounts of motion blur within the short integration time (e.g., 6 milli-seconds).

### 2.3. Data Processing and Spatial Analysis Techniques

Most, if not all, of the data examined in this study has been ortho-rectified by 3rd party software that came standard with the HS VNIR instrument under test. This software will, of course, be in question for some of the reasoning behind the artifacts seen in the results; however, uncorrected hyperspectral imagery was assessed to provide confidence in the technique of extracting spatial misregistration from point targets. We observed that for multispectral imaging, multiple mirror targets produced a well-behaved SPSF and errors in ortho-rectified imagery of single mirror targets were mainly due to the instrument and not the software processing over localized regions on the focal plane. There are two procedures for estimating the SPSF, depending on the imaging modality and number of mirrors used in the estimation algorithm. A 2D Gaussian distribution is the main assumption used for modeling the spatial response of aliased imaging systems (i.e., detector-limited imaging). This function is used widely throughout the processing of both multispectral and hyperspectral imagery. However, only the cross-track direction will be discussed, as previously mentioned, for the HSI example.

For multispectral imaging, ortho-rectification of the scene is not required, and all 2D spatial information is captured during the quick integration time (e.g., 4 milli-second integration time). Without the worry of ortho-rectification, multiple mirror targets can be used to oversample the spatial response to form an estimate of the SPSF. The use of multiple mirrors provides the missing information lost when an aliased imaging system observes a scene. The sample phasing of the mirror targets (i.e., sub-pixel location of the point source relative to the square pixels) is taken advantage of in the algorithms proposed by Schiller and Silny [12,14]. On our work here, the algorithm (outline shown in Figure 8) was re-created in Python to perform a multi-point analysis where multiple mirror targets produced an aliased spatial response that can collectively estimate the imaging system SPSF under the assumption of a 2D Gaussian distribution. The double-loop algorithm [12] optimizes a pair of x and y Full-Width at Half Maximum (FWHM) estimates that will minimize the sum of the root mean square error (RMSE) from all target responses. The inner loop uses an initial guess of the x and y FWHM to estimate the center positions of all target responses, then calculates the resulting RMSE. The outer loop varies the directional FWHM for the inner loop and minimizes the sum of RMSEs that fit best with all the target responses. This minimized x and y FWHM defines the 2D Gaussian distribution that fits the collective data best. It will be shown that for aliased imaging systems, the 2D Gaussian distribution produces a well-behaved SPSF estimate for multispectral modalities.

For hyperspectral imaging, the luxury of using multiple points becomes more complex due to the uniqueness of the spatial response as a function of FOV and the errors that arise related to ortho-rectification. Thus, the mixture of multiple-point targets for any fixed mounted HSI systems cannot be fully justified (see Section 3). Using the 2D Gaussian distribution and a single point target, the first-order approximations to the spatial misregistration of any HSI systems can be calculated. The SPSF signals are fitted using a non-linear optimization function where wavelength-dependent FWHM and center positions are estimated, relieving the spatial misregistration within the HSI system under test, including any FOV dependencies. Only cross-track directional estimates will be the primary focus of this study since this can be tied directly to the instrument’s performance with little ortho-rectification dependencies.

Using this fitting routine, the information depicted in the third row of Figure 2 can be dissected into two important parameters defined in the previous two rows. Keystone can be estimated by analyzing the center position in thee cross-track direction as a function of wavelength. When HSI systems image point targets, all wavelengths are measured simultaneously, where displacements in the point target’s centroid position can be measured relative to a reference wavelength. In this analysis, all centroid positions are measured relative to the position at wavelength 700 nm because this is the midpoint of the spectral range. Since all point targets are represented by rows and columns with the origin at the top left (i.e., coordinate (0,0)), a positive displacement means the point target was displaced to the right, and a negative value means the point target was displaced to the left.

The SPSF width variability as a function of wavelength can also be assessed from the fitting routine. During the fitting routine, a strict lower limit on the FWHM of 1 pixel can be set based on the limiting blur factor of a single pixel size. For example, an infinitelyrst orrst order appr sharp optical PSF convolved with a rectangular pixel will merely blur the edges, and the resulting shape will still resemble a rectangular pixel. However, fixed mounted HSI systems do not strictly follow these rules because ortho-rectification can corrupt and reconstruct unrealistic SPSFs when motion is too complex during flight. This will be demonstrated with various data sets over point targets, where it will be shown that fix mounted HSI systems can undersample scenes that can cause further reconstruction inaccuracies.

Because the SPSF signal is modeled by a 2D Gaussian distribution, a metric for estimating the spatial misregistration (see Section 1) within HSI systems can be easily calculated using only cross-track parameters. Each wavelength has a corresponding 1D Gaussian distribution that is area normalized and is used for this first-order approximation to the coregistration error. A 2D mapping will be shown where each wavelength pair (i.e., (400, 401 nm), (400, 402 nm), …, (400, 1000 nm)) has a corresponding spatial coregistration error with an added colorbar. A histogram of the coregistration error for all wavelength pairs will also be displayed to analyze how the errors are distributed, along with an FOV-dependent observation to understand how they differ.

## 3. Results and Discussion

Defining a vicarious technique to assess the spatial misregistration is important for understanding the end-to-end spatial performance and application-based limitations of HSI systems. A quantitative analysis of the spatial response of an HSI system to a point source will be discussed with parameters specific to keystone and SPSF width variability. This evaluation will also provide a first order approximation to the spatial coregistration error and a means to evaluate any FOV dependence on the SPSF for cross-track performance only. As previously mentioned, cross-track analysis in pushbroom HSI systems ties directly back to internal performance. Before assessing HSI spatial performance, a multispectral imaging system with a similar IFOV will demonstrate the use of point targets without the need to ortho-rectify the imagery.

### 3.1. Multispectral Imaging Results—Example

To demonstrate the use of convex mirrors as ideal point targets for extracting an estimate of the SPSF, a simplified experiment was conducted using an MSI system. This experiment will provide confidence in extracting the SPSF when ortho-rectification is not an issue. More importantly, under the assumption that both systems are spatially aliased (i.e., detector-limited imaging), the multi-point analysis of the multispectral system will demonstrate the similarity between the Gaussian distribution and the extracted SPSF. With trust in using the Gaussian distribution for the HSI system, simplifying the experiment (i.e., using only one mirror for spatial performance) can be achieved.

With many observations of the SPSF, sampling artifacts such as aliasing can be mitigated and estimates of the SPSF’s unique shape can be extracted and compared with the 2D Gaussian distribution. For this experiment, only the blue channel was used to demonstrate the oversampling technique. The goal of this analysis is to provide reasonable confidence that a 2D Gaussian distribution models an aliased imaging system without the need for ortho-rectification. Figure 9 illustrates our results. Figure 9a shows 16 point targets that were used to build a non-uniformly spaced SPSF. Figure 9b is a 3D representation of the SPSF with the optimized 2D Gaussian distribution.

To ensure the results focused on the SPSF peak, the image chips were constrained to 7 × 7 pixels and allowed for all enclosed energy to be captured. In addition, a flat-field image derived from an integrating sphere was applied to minimize any Photon-Response Non-Uniformity (PRNU) which slightly improves the RMSE due to the near Lambertian background (black felt). Table 2 contains all the results from this analysis. The major takeaways are the close to unity FWHM, which was expected, the low uncertainty in the FWHM (derived from the fitting routine), and the low RMSE. The small uncertainty in the FWHM can be attributed to the localized region where the point targets were deployed (i.e., only local aberrations and optical PSF) and the fact that the multiple point targets individually represent a collective surface that is the SPSF. A large variation in the FWHM uncertainty would indicate that the individual point responses are in fact different from one another and could not collectively form one SPSF. The low RMSE can be traced back to two factors: the background being uniform with low reflectance and the fact that the unobstructed aperture and detector-limited imaging system can be accurately modelled by a 2D Gaussian surface.

### 3.2. Estimation of Keystone—Hyperspectral

Keystone has significant impacts on spatially dependent targets, especially when light from the target/background area is spectrally displaced into neighboring spatial pixels. This displacement mixes the spectral components of the target/background such that the resultant spectrum will be non-ideal. More importantly, we will show that keystone error will have FOV dependence such that there is a varying trend across the focal plane, further complicating the issue of spatially separating targets. Figure 10 demonstrates the off-axis behavior of the HSI system under test as the point source moves towards either edge of the focal plane. What can be seen in Figure 10 is that there is very little keystone in the center FOV compared with either edge of the focal plane. These results highlight the problem areas to avoid (i.e., the edge of the focal plane), but also demonstrates the technique’s sensitivity in extracting the unique properties of HSI systems.

False-colored images (B-500, G-700, and R-900 nm) were used for displaying the three point targets at the top of Figure 10. Since the solar spectrum peaks around 500 nm, a well aligned system will exhibit a blue tint in the false-colored images. Based on the HSI system’s spectral range (400–1000 nm), a reference wavelength of 700 nm was selected to show how the SPSF’s relative centroid position varies as a function of wavelength. Major color separation can be seen in the false-colored point target images (at the edges of the FOV), which are reflected by the “S” shaped curves in the plot below the images. A positive value in the graph defines a shift to the right, whereas a negative value defines a shift to the left with respect to 700 nm. An increasing consequence of this analysis is that either side of the focal plane acts oppositely and uniquely, highlighting the usability of mirrors for a vicarious assessment of keystone. Ideally, any imaged point target should not wander about the reference wavelength since this would yield evidence of spatial pixel corruption.

The result at the center part of the FOV illustrates the best scenario since energy is separating minimally from surrounding spatial pixels. This is further highlighted by the overall spread of the SPSF in the images at the top of the graph. Since the fore-optic was more accessible than the internal spectrograph components, the fore-optic data sheet revealed that this particular objective lens suffers from barrel distortion. Objective lenses that suffer from barrel distortion have better on-axis performance than off-axis. This further explains the results seen in Figure 10. Depending on the internal spectrograph design, one can speculate that spherical and/or parabolic mirrors are used to relay the image to the focal plane. These designs can further complicate off-axis behavior [17].

Since a pushbroom instrument collects 2D spatial information by forward motion, small regions on the focal plane can be assessed to determine the consistency of the cross-track keystone for well-behaved SPSFs. Figure 11 shows the SPSF keystone between three point targets, all on the right side of the FOV. Even though this is a different data set with unique motion than the data set used in Figure 10, similarities can be easily observed. The slight differences in the cross-track keystone shape seen between Figure 10 and Figure 11 can be attributed to the slight differences where the point targets land on the right side of the FOV. Even with these subtle differences, the consistency in extracting keystone within a localized region and overall trends further demonstrate the uniqueness in using point targets as a method for extracting critical information about instruments used during field experiments.

All three curves trend collectively and demonstrate the repeatability even though the overall shape of the SPSF’s look different. Moreover, the ortho-rectification directly impacts the shape of the SPSF’s adding error in the cross-track fitting. However, these impacts are inherent in fix mounted HSI systems, where all pixels within a few frames will experience similar behavior. Further investigation needs to be completed on the discrepancies in the along-track center position seen in Figure 11. Since the point sources were offset in the direction of flight, weather impacts and ortho-rectification errors may be deduced from this type of data. An expectation of the along-track center position not being zero and spectrally flat could indicate overall flight and image reconstruction.

### 3.3. Estimation of SPSF Width Variability—Hyperspectral

The SPSF width is the other important parameter for understanding misregistration in HSI systems. More specifically, for any imaging system, the SPSF width will never be constant due to the nature of diffraction. Moreover, with perfect optics, SPSF width variability will be at best linear and increasing at longer wavelengths (i.e., the Rayleigh criterion). In most HS instrument designs, there tends to be an emphasis on high signal-to-noise ratio (SNR) and spectral separability. Through the analysis of the multispectral imaging system (Section 3.1) and the following discussion, these systems exhibit detector-limited imaging where the SPSF’s FWHM is close to unity.

Figure 12 shows the results from extracting the SPSF FWHM at three locations across the focal plane. These are the same point targets from Section 3.2. The FOV center resembles what would be expected for a detector-limited imaging system, where the SPSF’s FWHM is close to unity and grows towards 1.5 pixels at longer wavelengths. Noise dominates the estimate of FWHM on either side of the spectral range, but the overall trend demonstrates the SPSF’s width variability at the center FOV will dictate spatial resolvability and spectral separability of spatial targets.

When the off-axis point targets are estimated, interesting features can be seen throughout the curves, with similarities and differences between each other. Both off-axis SPSF’s have an interesting increase in FWHM around 700 nm, but deviate from each other below 650 nm. The one inconsistency seen throughout all SPSF is within the left FOV FWHM estimate, which dips below 1 pixel. This is physically impossible due to the limiting blur factor of the pixel size (i.e., FWHM cannot be less than 1 pixel) and can be attributed to ortho-rectification errors and an unrealistic reconstruction of the instrument SPSF. This will be further discussed in Section 3.5.

The assessment of the FWHM in the cross-track direction as a function of wavelength can highlight the unique behavior of spatial performance and impact scientific applications such as the spectral unmixing of sub-pixel targets. Figure 13 shows the consistency of the cross-track FWHM for two point targets in close proximity with good ortho-rectification. It is important to note that the point target in the orange box is a 2-mirror array aligned in the along-track direction. This has the potential of blurring the along-track FWHM but is still considered a sub-pixel target in the cross-track direction. This result demonstrates the consistency of the SPSF over a small localized region on the focal plane that can be extracted from this technique. These curves trend together between 400 and 800 nm, then slightly deviate after 800 nm due to errors in the fitting procedure and a lack of contrast between the background/target signals.

An along-track analysis will not be shown here and will be discussed further in Section 3.5. The fixed mounted HSI system has no active damping, which can drastically manipulate the shape and size of the SPSF in the along-track direction. Furthermore, this can cause inconsistencies in the reconstruction of the SPSF. Moreover, Figure 4a contains a 2-mirror array aligned in the along-track direction, and since the separation between these point sources is 5 cm (image GSD is 6.5 cm), this will elongate the along-track FWHM.

### 3.4. Estimation of Spatial Coregistration Error—Hyperspectral

As previously discussed in Section 1, spatial coregistration error is a comparative estimate of the shape, size, and location of the SPSF for all wavelength pairs and provides a single metric for HSI spatial misregistration. In this study, only two SPSFs were analyzed (center FOV and the right FOV SPSF) to compare the FOV dependence on misregistration extracted from field experiments. This study was specifically designed to create a fast and simple approximation to estimating the spatial coregistration error using only a single SPSF response and a Gaussian fit. This approximation can highlight the size and location related to spatial misregistration where the unique shape will be (somewhat) lost because only the optimized Gaussian distribution is used. The unique shape is an important parameter since any asymmetry could have a dramatic effect on the spatial coregistration error; however, non-aliased SPSF’s require a multi-point analysis, which can be troublesome with fixed mounted HSI systems.

Figure 14 shows an example of quantifying the HSI misregistration for the HSI system under study by calculating the spatial coregistration error. The center and right FOV SPSFs from Section 3.2 and Section 3.3 were used to demonstrate how the keystone and FWHM estimates come together into a wavelength pair metric for a more simplistic look at spatial misregistration. The spatial coregistration error was calculated using the FWHM and centroid position (cross-track only) derived from the Gaussian fitting routine, and both were smoothed consistently to lower dependencies on the noise at the longer wavelengths.

Quantifying spatial misregistration (Figure 14) provides a wealth of knowledge about the end-to-end performance of the HSI system by characterizing how well a point source is deposited onto the focal plane and is directly linked to the instrument’s spatial performance. These single metrics provide a user with a fast and simple calculation to assess how the HSI instrument focuses light originating from a sub-pixel point source. The small image chips in Figure 14 show the false-colored image (B-500, G-700, and R-900 nm) for visually understanding why the right FOV is worse than the center FOV. The overall color separation is indicative of misregistration, and the neighboring spatial pixels are corrupted more at the right FOV than at the center FOV.

Initial observations of the similarly scaled heat maps in Figure 14 clearly highlight optimal performance of the HSI system at the center part of the FOV. Spatial coregistration error calculated from single point targets clearly demonstrates the uniqueness of this technique for assessing spatial misregistration during field experiments. Small targets are greatly affected by spatial misregistration because optical features such as keystone distort their spectra unrealistically and make it more difficult to compare with materials extracted from spectral libraries or, in this case, extracted from differing parts of the FOV. This technique provides a user-friendly solution to assess trouble areas of the FOV where optical features could corrupt application performance. Skauli (2012) provides the initial theoretical framework for including spatial coregistration error into an example involving the difference between two spectra and how this can influence the perceived spectral signature. The use of spatial coregistration error in applications can provide additional thresholds that are unique to the instrument’s capability to spatially resolve spectral targets from background or cluttered areas.

There are many important features in the spatial coregistration heat maps and histograms that require further discussion. The main takeaway is that the ideal performance of the HSI system is on-axis based on the low average spatial coregistration error, where the spatial width is the limiting factor. Performance will smoothly vary and gradually get worse as one images closer to the edges of the FOV, as demonstrated. The apparent bounding box seen in the Right FOV heat map example is entirely due to keystone dominating the spatial coregistration error calculation (to be further explored in Figure 15).

To further highlight features within the spatial coregistration error example (i.e., Figure 14), example plots of Gaussian distributions used in the error calculation (Equation (Equation 1)) for wavelength pair (500, 900 nm) are seen in Figure 15. For the reason the Center FOV example contains differing spatial misregistration than the Right FOV example, Figure 15 also highlights why large spatial coregistration error is experienced at the edge of the focal plane.

Illustrated in Figure 15 are the area normalized Gaussian distributions for wavelength pairs of 500 nm and 900 nm derived using the parameters estimated in Figure 10 and Figure 12. Since the Center FOV example contains mainly spatial width dependence, the distributions are overlapping and only vary in height and width (i.e., SPSF at 900 nm is wider than SPSF at 500 nm). Whereas in the Right FOV example, there is over 1 pixel of keystone, but both distributions have similar spatial widths. If we observe Equation (Equation 1), the spatial coregistration error is a difference between area normalized SPSFs (i.e., in this case, Gaussian distributions fitted to point targets). By observation, the disjointed SPSF’s Figure 15b of similar width will lead to a larger error than the overlapping SPSFs Figure 15a of varying width.

### 3.5. Ortho-Rectification Impacts—Hyperspectral

As with all pushbroom HSI systems, ortho-rectification is required to reconstruct the observed scene. This process is extremely dependent on the stability of the airborne platform, GPS/IMU accuracy, and Digital Elevation Mapping (DEM). Unless the HS instrument is actively damped with a gimbal mount, the ground targets of interest can be oversampled or undersampled based on the drone’s pitch and roll. The results in this section will highlight observations made on point targets that were not correctly reconstructed and provide further insight on what a pushbroom HSI system is observing. Using point targets (i.e., convex mirrors) provides an easy technique to assess ortho-rectification impacts in all imagery collected and has the potential to highlight the loss of scene content.

Before displaying corrupted imagery, a short study on the impacts of ortho-rectification on a well-behaved point target will be assessed. Figure 16 represents the comparison between a raw, uncorrected image and its ortho-rectified version. Due to the fixed mounting of the HSI system under test, this is not always possible because, as will be discussed at the end of this section, the imaging system can roll and/or pitch aggressively as it images point targets, which corrupts the target in the unortho-rectified imagery. Figure 16 demonstrates that the ortho-rectification process can preserve the instruments spatial performance for well-behaved SPSF’s. There is slight deviations at the lower wavelength range that can originate from a tighter SPSF being more susceptible to motion throughout the imaging process than a wider SPSF.

To show the inconsistent ortho-rectification, due to complex motion blur, a side study was conducted with two similar point sources imaged multiple times within a small time period, as seen in Figure 17. What this image shows are two separate flight lines over the same target with dramatic differences in shape, size, and extent of the SPSF. Chaotic pitching of the drone can under or oversample the scene, causing severe smearing and issues in the ortho-rectification that cannot be completely fixed without active damping. These issues are always present in fixed mounted HSI systems and will impact the spatial/spectral separability of in-scene targets.

For fix mounted HSI systems, motion can cause scenes to be under- or oversampled and can be easily observed when imaging sub-pixel spatial targets (i.e., convex mirrors). Ortho-rectification is always required to create 2D square pixel imagery, but this can be misleading because ground coverage is highly dependent on frame rate and motion between frames. Missing ground coverage requires interpolation, and without additional information, this is, at best, guesswork when creating the final HS imagery. Figure 18 shows an example of undersampling the image scene that was highlighted in Figure 4. The three main targets were not sampled properly due to aggressive pitching and rolling over that area (Figure 18a) where the targets seemed squashed and the targets in the orange box only have a single row of data. The ortho-rectified image (Figure 18b) contains about 2× more pixel data over the same area with the targets inside the orange box being elongated in the cross and along-track direction.

Ortho-rectifying undersampled scenes inherently requires interpolation to recover the missing area in the along-track direction. The recovered pixels, therefore, cannot be directly linked to the scene’s physical quantities due to the lack of observed photons. This has major implications when the imagery is used for scientific applications such as small target detection or sub-pixel spectral unmixing. The interpolation process alone cannot recover the lost information, and these artifacts can be hidden without sub-pixel calibration targets (i.e., convex mirrors). The radiometric and spatial interpretation of this imagery cannot be trusted, and the point targets described in this paper provide a method to understand this process due to their sub-pixel nature. As more applications move towards sub-pixel detection, it is critical that these artifacts are understood, and the only method for doing this is using the presented convex mirrors.

Oversampling the image scene presents the other end of motion-induced artifacts. This issue is not as damaging to the underlying physics because there is an excess of information or duplication that requires reduction. The problem with too much information is that the ortho-rectification process has to deal with reducing the imagery without over-fitting. Figure 19 shows an example of when point targets are oversampled and is highlighted by the orange boxes. The small point targets begin to resemble line sources in the raw imagery because the change in pitch/roll angle synchronizes with the forward motion of the drone. Furthermore, the point targets in the raw imagery highlight how a sub-pixel target would be reconstructed in the ortho-rectified image. Thus, the method provides for quality checks over targets of interest.

For any fixed mounted HSI systems, motion will plague the image scene with the potential of misrepresenting the physical properties to be observed. The point targets provide a method for understanding how the raw imagery was collected for quality assurance checks. For the reason that these mirrors represent a sub-pixel target having absolute radiometric and spatial properties, there is a potential to unitize point targets as a vicarious technique for diagnosing ortho-rectification errors over an area containing scientific targets of interest. Future research efforts are focused on using the unique capability of convex mirrors, which have a known spectral radiance signature, as a technique to address ortho-rectification errors.

## 4. Conclusions

Estimating the spatial performance of an HSI system requires extensive laboratory equipment, including collimators, bright illumination sources, and high-accuracy translation stages. The SPARC technique was proposed for estimating the spatial performance of HSI and MSI systems during field experiments without an expensive laboratory setup. In this study, the novel approach to assessing hyperspectral misregistration, using mirror targets, was demonstrated during UAV field experiments of varying altitudes and GSD’s. It was shown that spatial parameters such as SPSF width and keystone that contribute to spatial misregistration, can be easily extracted from field data using only a single mirror target. Not found in previous literature is a vicarious technique to bridge the gap between laboratory and field-based HSI performance testing and comparison, with a focus on extracting hyperspectral spatial misregistration. Diagnostic testing, spatial validation, and FOV dependencies are an added benefit to the deployability of point targets during field experiments. In addition, this technique can provide vital information about the ortho-rectification process and its limitations for scientific applications requiring sub-pixel detection methods. An awareness as to how imaging systems perform during field experiments is critical for advancing the understanding of collected data quality for all scientific applications. The experiments and analysis presented in this paper are also directly applicable to airborne or satellite-based imaging platforms where point targets provide a sub-pixel spatial response.

## Figures and Tables

**Figure 1 sensors-23-04333-f001:**
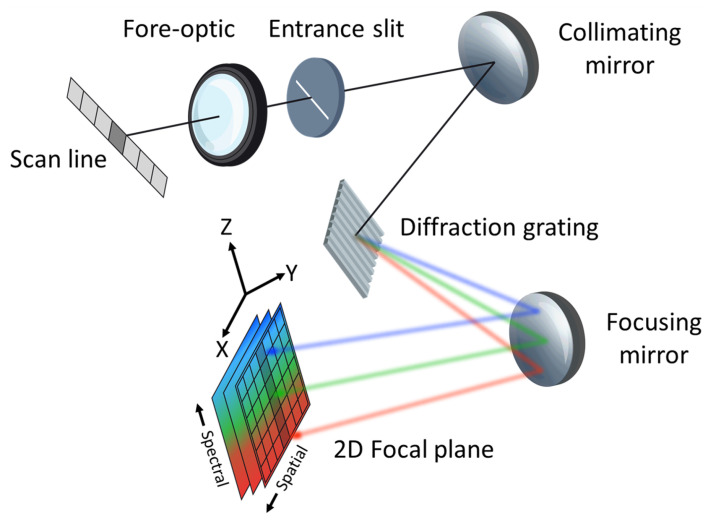
An illustration of a hyperspectral instrument highlighting the various components resulting in a spatial-spectral image of the scan line at the focal plane. The *x* (cross-track) and *z*-axis define the internal spectrograph orientation where the scan line is dispersed into a spectrum. The *y*-axis (along-track) defines the second spatial dimension collected when the instrument is pushed forward.

**Figure 2 sensors-23-04333-f002:**
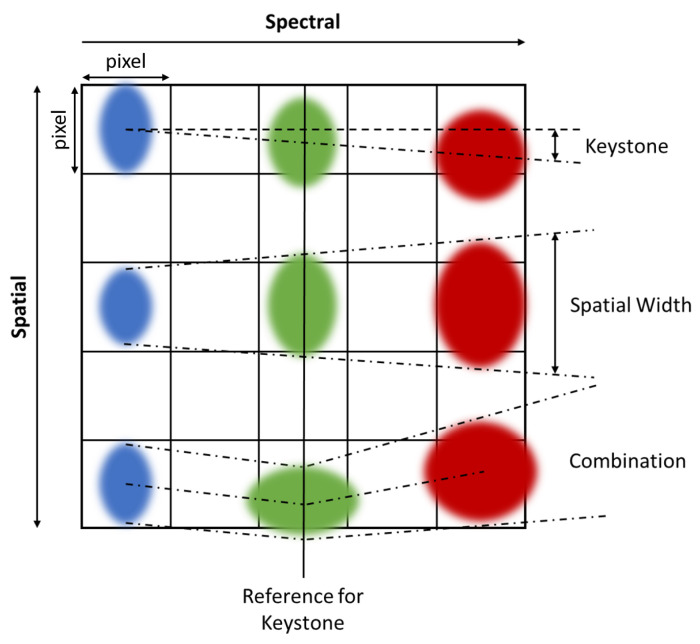
Examples of spatial distortions seen in typical HSI systems. Top row is an example of keystone. Middle row illustrates distortions due to diffraction and aberrations while the bottom row shows a combination of both distortions types.

**Figure 3 sensors-23-04333-f003:**
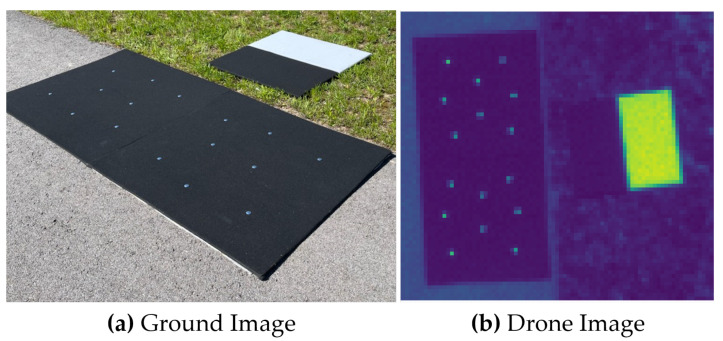
(**a**) Ground image of the point and edge targets. (**b**) Resulting image of the targets from the MSI system.

**Figure 4 sensors-23-04333-f004:**
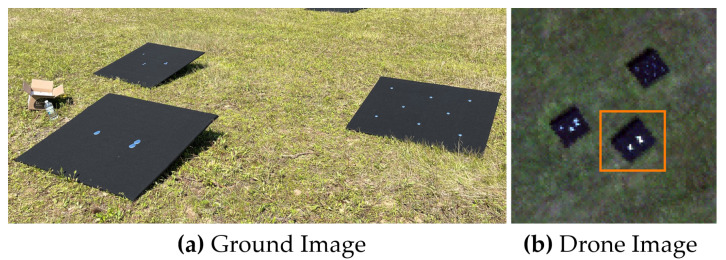
(**a**) Primary site for G-SCALE where point targets were deployed for radiometric and spatial testing. (**b**) HSI image (RGB bands shown) over the point targets where the orange box highlights the targets used in the analysis.

**Figure 5 sensors-23-04333-f005:**
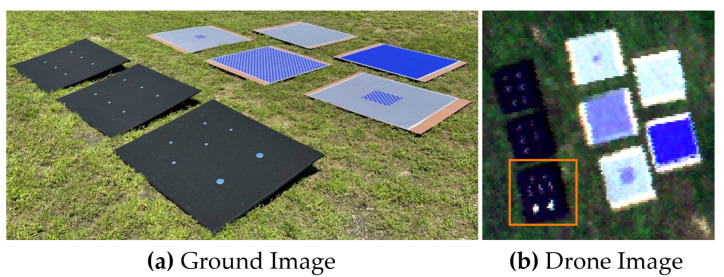
(**a**) Secondary site for G-SCALE where point and spectral unmixing targets were deployed for additional spatial testing. (**b**) HSI image (RGB bands shown) over the point and spectral unmixing targets where the orange box highlights the targets used in the analysis.

**Figure 6 sensors-23-04333-f006:**
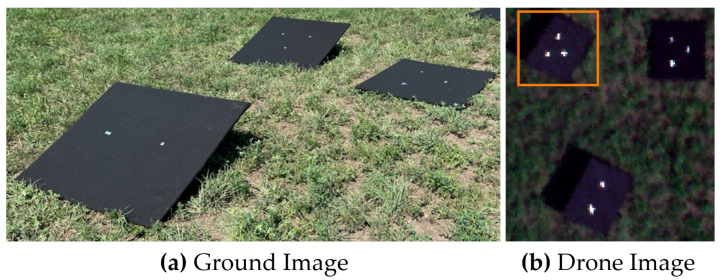
(**a**) Main site for BigMAC where the primary extraction of surface reflectance for HSI calibration was tested. (**b**) HSI image (RGB bands shown) over the point targets where the orange box highlights the single mirror arrays used in the analysis.

**Figure 7 sensors-23-04333-f007:**
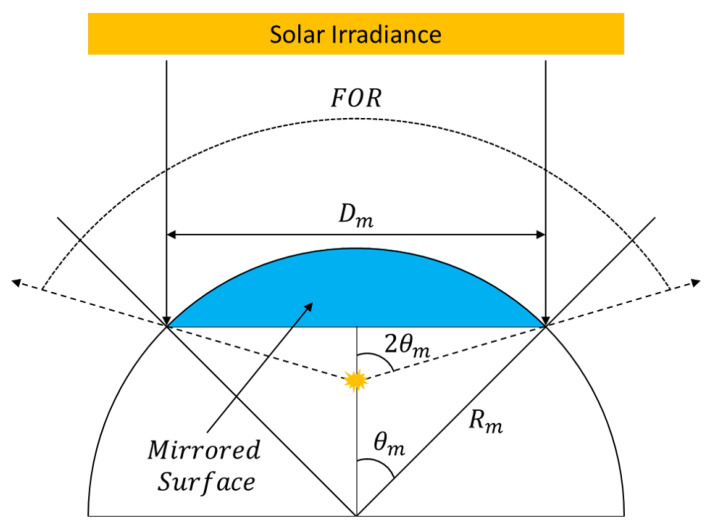
Various geometrical properties of convex mirrors when stimulated by plane waves originating from the sun.

**Figure 8 sensors-23-04333-f008:**
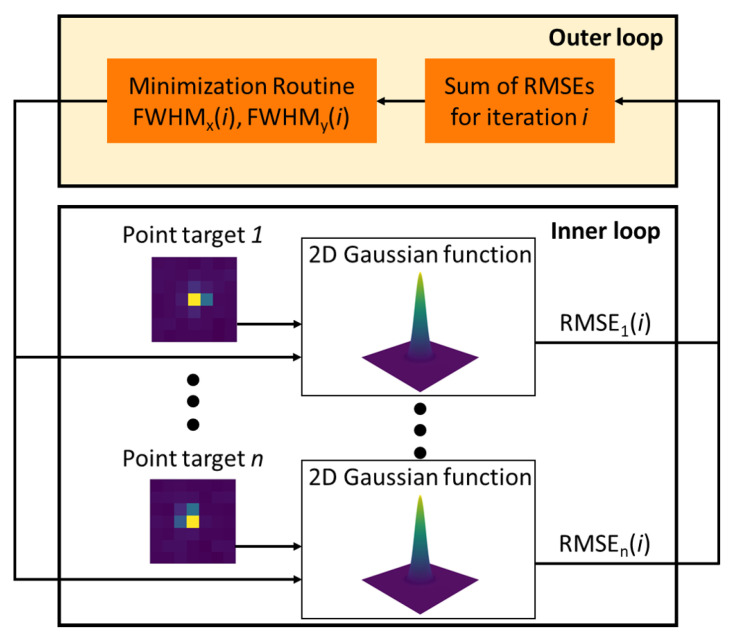
The double-loop algorithm estimates the x and y FWHM by minimizing the sum of the RMSE’s from all target responses. The inner loop estimates the targets center position with a fixed x and y FWHM and the outer loop optimizes the directional FWHM until a solution is found.

**Figure 9 sensors-23-04333-f009:**
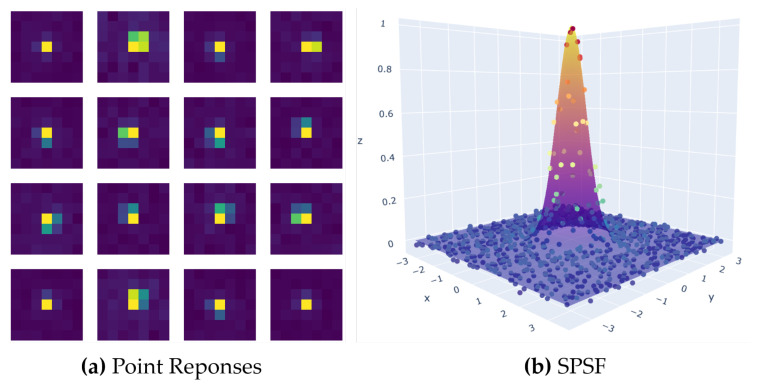
(**a**) The collection of phased point responses used in the creation of the SPSF. (**b**) Final SPSF result when using a 2D Gaussian distribution to centroid multiple point targets into a common reference frame.

**Figure 10 sensors-23-04333-f010:**
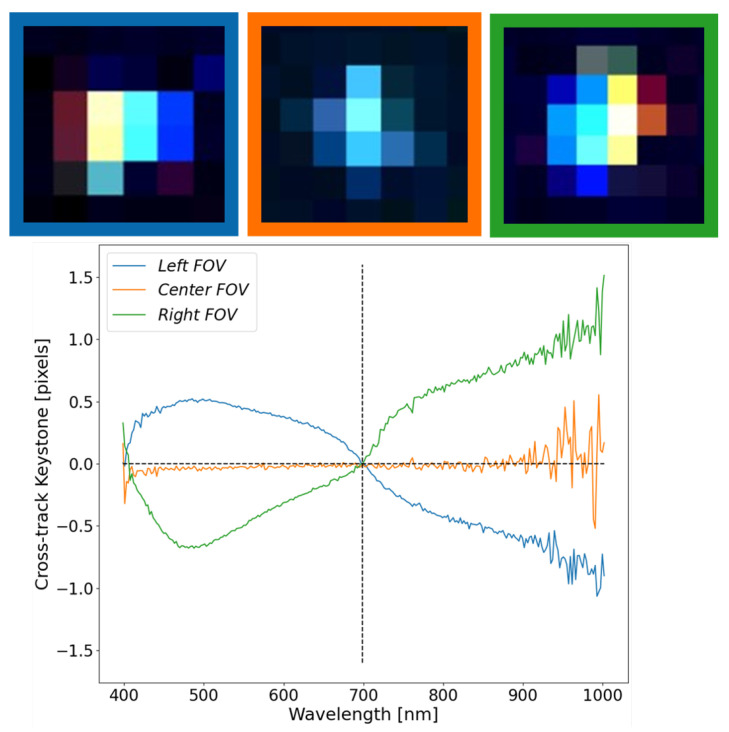
FOV dependency in the cross-track center location (keystone) for three positions of the focal plane. The colored boxes around each SPSF link to the colored curves in the graph below.

**Figure 11 sensors-23-04333-f011:**
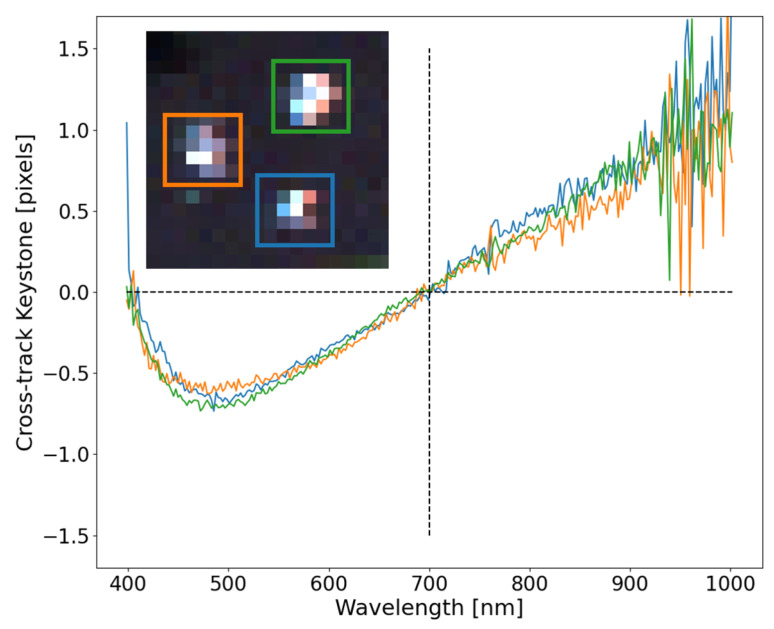
Consistency for the three localized point targets demonstrate the robustness in extracting a vicarious estimate of keystone. Slight ortho-rectification errors can be seen and color separation trends with previous experiments.

**Figure 12 sensors-23-04333-f012:**
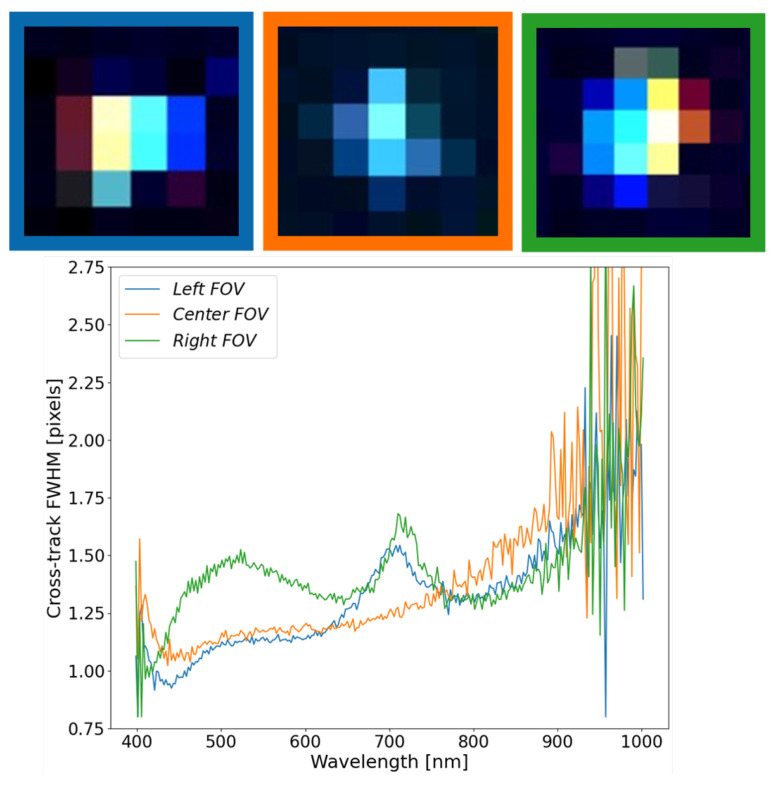
The FWHM of SPSF’s at at the FOV edges and center FOV demonstrates uniqueness of the shape and size across the FOV. The SPSF’s FWHM at either edge has interesting features around 700 nm, but tend to show that this HSI system is detector-limited with FWHM values close to unity.

**Figure 13 sensors-23-04333-f013:**
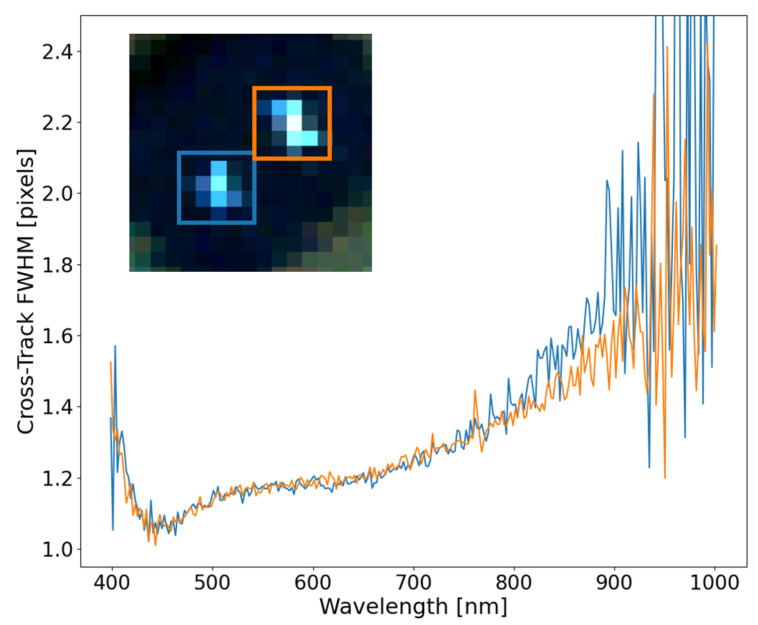
With good ortho-rectification, close point targets exhibit similar FWHM behavior in the cross-track direction and can be tied back to the instruments spatial performance. The colored boxes connect to the curves.

**Figure 14 sensors-23-04333-f014:**
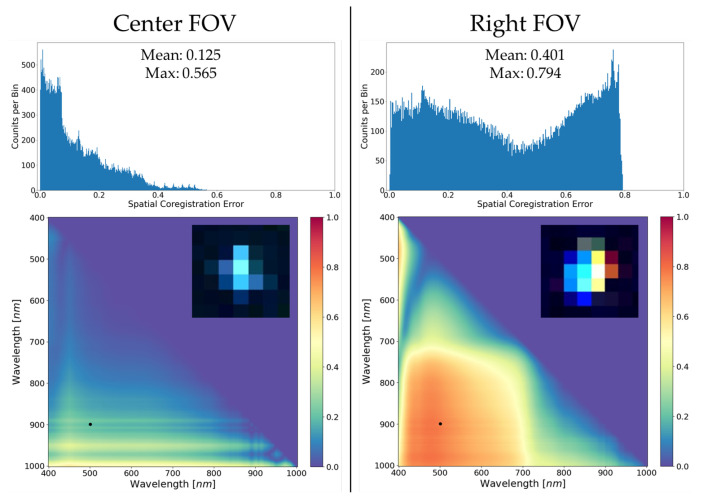
Examples of spatial coregistration error and how off-axis effects greatly degrade the image resolution and separability of targets from background in the area surrounding the SPSF’s. The histograms summarize the range and spread of the spatial coregistration errors with mean and max values. The heat maps at the bottom are a visual representation of spatial coregistration error for all wavelength pairs and is mirrored about the y=x axis (i.e., wavelength pair (400, 401 nm) is equivalent to (401, 400 nm)). Black dots within the heat maps represent a specific wavelength pair to be discussed.

**Figure 15 sensors-23-04333-f015:**
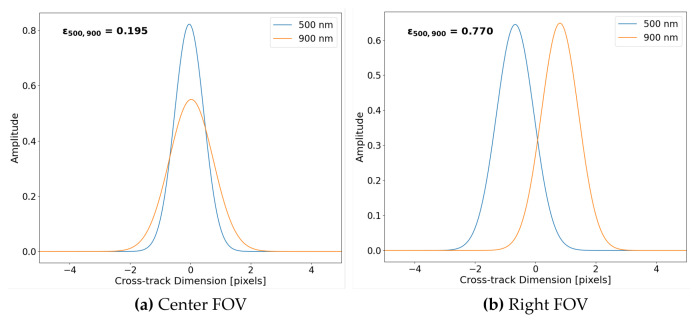
Two examples of area normalized Gaussian distributions for wavelength pair (500, 900 nm) extracted from the spatial coregistration error example with their respective error values in bold text. (**a**) Due to small keystone, the leading contributor to spatial misregistration is FWHM differences. (**b**) Large keystone error contributes greatly to the spatial coregistration error for wavelength pairs with similar FWHM.

**Figure 16 sensors-23-04333-f016:**
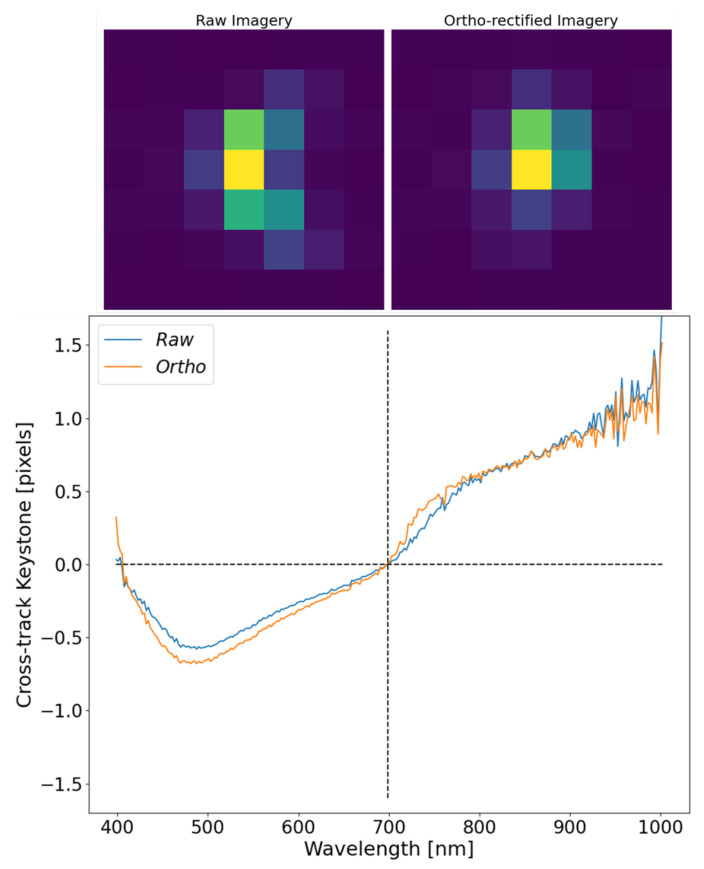
The top of the graph shows images of the same SPSF (raw and ortho-rectified) displayed at a single wavelength of 550 nm. The results from estimating keystone with respect to 700 nm show how minimal impacts from ortho-rectification affect the displacement of the SPSF as a function of wavelength.

**Figure 17 sensors-23-04333-f017:**
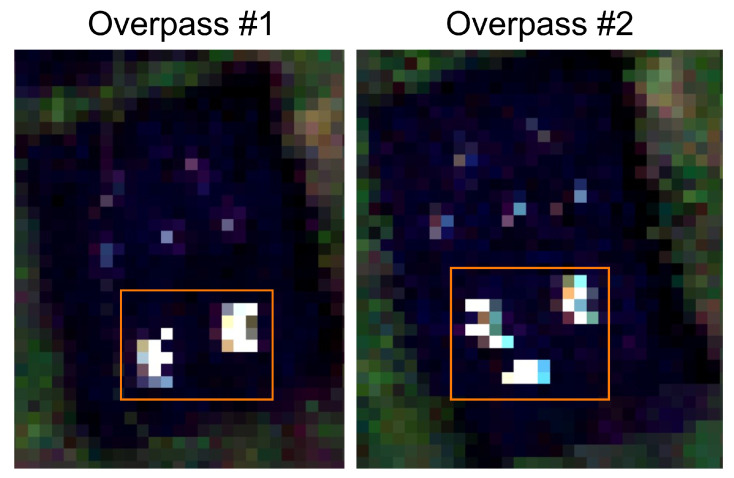
Inconsistent ortho-rectification over the same point targets from two different flight lines. The orange boxes highlight the targets of interest and only two mirrors were present, but Overpass #2 falsely contains three. Reprinted/Adapted with permission from Ref. [10]. © 2022 IEEE.

**Figure 18 sensors-23-04333-f018:**
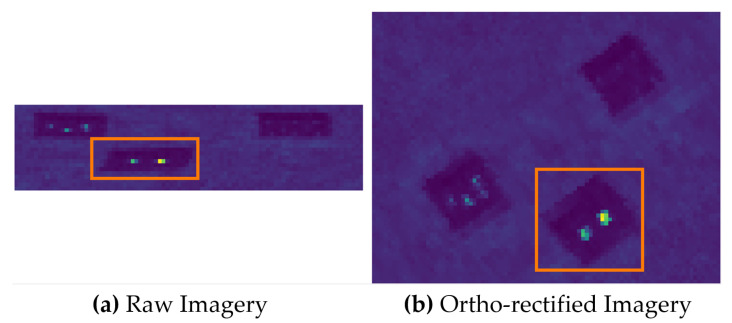
A comparison between raw and ortho-rectified imagery for a scene that was undersampled due to the motion of the drone and fix mounted setup of the HSI system. (**a**) Raw imagery showing the undersampled area in the orange box. (**b**) The resulting ortho-rectified image of the same area.

**Figure 19 sensors-23-04333-f019:**
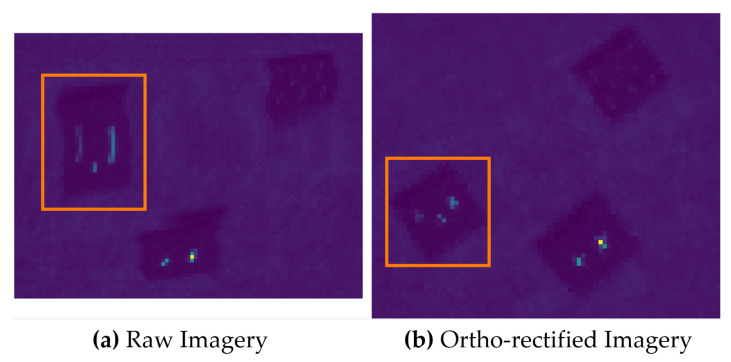
An example of a small area over the point targets that was oversampled by the drone motion. (**a**) Raw imagery with a small area that was oversampled highlighted in the orange box. (**b**) The ortho-rectified version with the orange box highlighting the same target area.

**Table 1 sensors-23-04333-t001:** Mirror configurations used in all field experiments. Various geometrical parameters highlighting the differing properties.

$R_{m}$	$D_{m}$	FOR	$d_{\mathrm{sun}}$
25 mm	22.9 mm	108.8^o^	0.11 mm
50 mm	22.9 mm	52.9^o^	0.22 mm
100 mm	45.7 mm	52.9^o^	0.44 mm

**Table 2 sensors-23-04333-t002:** Results from estimating the spatial performance of a multispectral imaging system using multiple point targets.

FWHM(Cross−Track)	FWHM(Along−Track)	RMSE
1.09 ± 0.57%	1.12 ± 0.57%	1.34%

## Data Availability

All relevant data can be requested by contacting corresponding authors.

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
