# Peer review of "A Vicarious Technique for Understanding and Diagnosing Hyperspectral Spatial Misregistration"

_sensors, 2023, doi:10.3390/s23094333_

Round 1

Reviewer 2 Report

This manuscript is designed to demonstrate a vicarious technique using convex mirrors to bridge the gap between laboratory and field-based HSI performance testing. This work has good practicality. But there is still room for improvement in the innovation of the abstract. Some adjustments to the manuscript are needed. The following are the primary issues I think should be addressed:

(1) Condense the abstract while highlighting the innovation points

(2) As in Figure 15 and Figure 16, when multiple diagrams are described together, it is recommended to name the different diagrams as (a), (b), (c), etc. instead of left and right diagrams.

Reviewer 3 Report

The necessary revisions and comments for authors;

I have re-read and re-evaluated the paper entitled as “A Vicarious Technique for Understanding and Diagnosing Hyperspectral Spatial Misregistration”. 

There are some major to minor concerns and comments about the submitted article which listed as follows;

I have some comments for the revised version of the manuscript;

1-      In figure 1 which is from Dell’Endice et. al. 2009, should be changed little bit. This figure shown as in original paper. Z-axis of the colored output shows spectral. However, spectral axis should be x-axis due to different spectral bands. Please check it. If the dispersing element exist to split the spectral channels then the number of plates could not be understandable. If possible you changed this figure for better presentation of the concept.

2-      Experimental results strengthen the idea of the proposed approach. Therefore, they should be clearly presented. Explanation is given in the text, but it is better to put scale-bar in figure 2 (right). Similarly, I recommend you to use scale bar for all images ant it is necessary to clear explanation.

3-      This study was performed in ground base hyperspectral cameras. Drones and aerial based images can also have or use similar hyperspectral cameras. But there is no any information about the flight height or capture distance in the text. It is better to give different images on same targets with same camera and different altitudes.

4-      It is very dense study with good presentation. It is better to improve the conclusion part to finalize this presentation. In addition also it is better to emphasize the novelty of the study in the text.
